# Efficient Construction of Model Family through Progressive Training Using Model Expansion

**Kazuki Yano[†], Sho Takase[†,‡], Sosuke Kobayashi[†], Shun Kiyono[‡], Jun Suzuki[†].**
[†]Tohoku University
[‡]SB Intuitions
yano.kazuki@dc.tohoku.ac.jp

## Abstract

As Large Language Models (LLMs) gain widespread practical application, offering model families with varying parameter sizes has become standard practice to accommodate diverse computational requirements. Traditionally, each model in the family is trained independently, incurring computational costs that scale additively with the number of models. In this work, we propose an efficient method for constructing model families via progressive training, where smaller models are incrementally expanded to larger sizes to create a complete model family. Through extensive experiments on a model family ranging from 1B to 8B parameters, we show that our approach reduces total computational cost by approximately 25% while maintaining comparable performance to independently trained models. Moreover, by strategically adjusting the maximum learning rate based on model size, our method outperforms the independent training across various metrics. Beyond these improvements, our approach also fosters greater consistency in behavior across model sizes.

## 1 Introduction

As Large Language Models (LLMs) gain widespread practical application, providing models with (i) a consistent architecture and (ii) varying parameter sizes (hereafter referred to as a **model family**) has become standard practice in the NLP community. For instance, Llama 3.1 includes models with 8B, 70B, and 405B parameters (AI@Meta, 2024), while Gemma 3 provides 1B, 4B, 12B, and 27B parameter variants (GemmaTeam, 2025) . Similarly, Qwen2.5 offers a model family with 0.5B, 1.5B, 3B, 7B, 14B, 32B, and 72B parameters (Yang et al., 2024b).

Such model families are designed to address a wide range of computational constraints and application scenarios. Smaller models offer faster inference and lower resource consumption, making them suitable for daily tasks and deployment in resource-constrained environments such as smartphones and edge devices (Abdin et al., 2024). In contrast, larger models are deployed for scenarios requiring advanced reasoning capabilities and complex task processing, typically on large-scale servers (Wei et al., 2022).

The standard approach to constructing a model family involves training each model independently from scratch. However, training large-scale models demands extensive resources, e.g., thousands of GPU days (Touvron et al., 2023). The total computational cost of constructing a model family poses a significant burden on its builders. (Figure 1 (*Top*)). This motivates us to explore more efficient methods for model family construction.

We identify **model expansion** as a potential approach to more efficiently constructing model families, including large-scale models. Model expansion leverages the parameters of pretrained smaller models as initialization for training larger models (Chen et al., 2022; Du et al., 2024). However, prior work on model expansion has primarily aimed at producing a single final model, with limited focus on the potential utility of intermediate models.

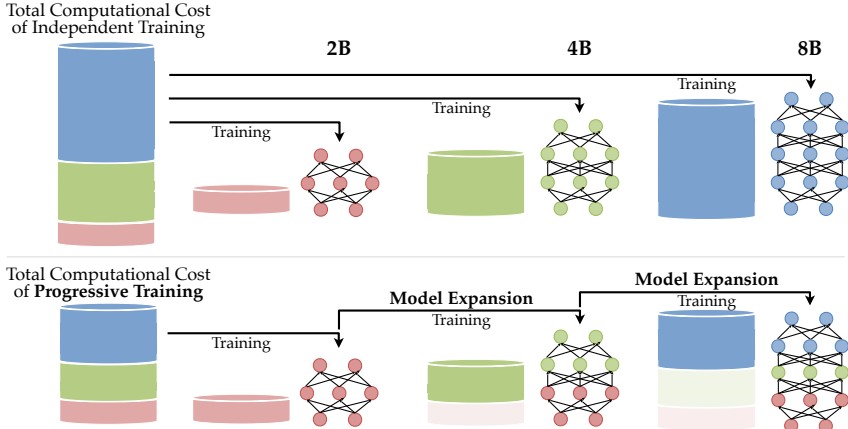

Figure 1: Comparison of approaches for constructing a model family. (*Top*): Conventional approach, where each model in the family (2B, 4B, 8B) is trained independently from scratch. The total computational cost is the sum of the training costs for all individual models. (*Bottom*): Proposed progressive training utilizes model expansion, where smaller models are expanded to initialize larger ones. The total cost equals only that of the largest model (8B).

In this work, we empirically explore model expansion as a means to efficiently construct a model family. Specifically, we propose a method that repeatedly applies model expansion to construct models from smaller to larger sizes, namely **progressive training**, thereby reducing the total training cost of constructing a model family (Figure 1 (*Bottom*)).

Through our experiments with a model family ranging from 1B to 8B parameters, we demonstrate that the proposed method reduces the total computational cost of constructing a complete model family by approximately 25% compared to training each model independently[1]. Moreover, by adjusting the maximum learning rate based on model size, our method consistently achieves superior performance across a range of benchmark tasks relative to independently trained models. We also found that progressive training yields greater behavioral consistency across the model family, as indicated by lower Kullback-Leibler (KL) divergence between models' output distributions.

## 2 Task Definition and Notation Rules

We consider a model family consisting of models with monotonically increasing sizes and consistent architecture. Formally, we define this model family as a sequence of models of sizes $[X_1, X_2, \ldots, X_n]$ and corresponding sequence of model parameters $[\theta_1, \theta_2, \ldots, \theta_n]$, where $\theta_i \in \mathbb{R}^{X_i}$ denotes the parameters of the $i$-th model. For each model of size $X_i$, let $T_i$ represent the number of training tokens used to train the corresponding model. The computational cost (FLOPs) required to train a given model is determined by the model size $X_i$ and the number of tokens $T_i$. Following Brown et al. (2020), we define FLOPs as: $\text{FLOPs}(X_i, T_i) = 6X_i T_i$. Note that our analysis does not depend on this specific formulation and can be generalized to other reasonable cost approximations (Hoffmann et al., 2022).

Let $T_i^{\text{scratch}}$ denote the number of tokens for training a randomly initialized model of size $X_i$ from scratch. The total computational cost of constructing the entire model family is then given by $\sum_{i=1}^{n} \text{FLOPs}(X_i, T_i^{\text{scratch}})$. The task of this work is to reduce the overall computational cost while preserving the performance of each model. Formally, we aim to

---

[1]This 25% reduction corresponds to savings of approximately 3.2K GPU hours in our experimental setup.

construct each model under the following constraint:

$$\sum_{i=1}^{n} \text{FLOPs}(X_i, T_i^{\text{scratch}}) > \sum_{i=1}^{n} \text{FLOPs}(X_i, T_i^{\text{prog}}), \tag{1}$$

where $T_i^{\text{prog}}$ denotes the number of tokens used to train the model of size $X_i$ under the progressive training approach (Section 3).

## 3 Method: Progressive Training

We propose a method for constructing language models of varying sizes more efficiently than the common practice, i.e., training each model independently from scratch. In the common approach, the cost of training a model of size $X_i$ from scratch using $T_i^{\text{scratch}}$ tokens[2] is $\text{FLOPs}(X_i, T_i^{\text{scratch}})$, leading to a total cost of $\sum_{i=1}^{n} \text{FLOPs}(X_i, T_i^{\text{scratch}})$ for the entire model family. A simple strategy to satisfy Equation (1) is to reduce the number of training tokens for each model by selecting $T_i^{\text{prog}}$ such that $T_i^{\text{scratch}} > T_i^{\text{prog}}$ for $i > 1$. However, naively reducing the number of training tokens typically results in degraded performance if other factors remain unchanged. Instead, we initialize the $i$-th model of size $X_i$ for $i > 1$ by using a better initialization than random, which is obtained by applying model expansion to the previously trained model sizesize $X_{i-1}$. We refer to this process as **progressive training**.

Specifically, our progressive training begins by training an initial model of size $X_1$ using $T_1^{\text{scratch}}$ tokens. At each subsequent stage, we initialize the next model using a model expansion method $f$,

$$\theta_{i+1}^{\text{init}} = f(\theta_i; X_{i+1}) \quad (i \geq 1), \tag{2}$$

where $f(\cdot; X_{i+1}) : \mathbb{R}^{X_i} \to \mathbb{R}^{X_{i+1}}$ is an off-the-shelf model expansion method. The expanded parameters $\theta_{i+1}^{\text{init}}$ serve as an effective initialization for training the subsequent model.

There exists a wide range of possible token allocation patterns $[T_1^{\text{prog}}, T_2^{\text{prog}}, ..., T_n^{\text{prog}}]$ that satisfy the constraint in Equation (1)'s condition regarding the total computational cost. In this work, we determine each $T_{i+1}^{\text{prog}}$ such that the overall computational cost matches that of training the largest model $X_n$ from scratch:

$$\text{FLOPs}(X_n, T_n^{\text{scratch}}) = \sum_{i=1}^{n} \text{FLOPs}(X_i, T_i^{\text{prog}}). \tag{3}$$

Specifically, we set $T_{i+1}^{\text{prog}}$ to satisfy the following FLOPs constraint:

$$\text{FLOPs}(X_{i+1}, T_{i+1}^{\text{prog}}) = \text{FLOPs}(X_{i+1}, T_{i+1}^{\text{scratch}}) - \sum_{j=1}^{i} \text{FLOPs}(X_j, T_j^{\text{prog}}). \tag{4}$$

Intuitively, we allocate the computational cost of training a model of size $X_{i+1}$ from scratch, subtracting the computational cost already used in previous stages. Through this procedure, we can obtain an entire model family from size $X_1$ to $X_n$ with a total computational cost equal $\text{FLOPs}(X_n, T_n^{\text{scratch}})$.

**How we choose $f$.** In this work, we adopt bert2BERT (Chen et al., 2022) as our model expansion method for the following reason. Unlike approaches that focus solely on depth expansion (such as stacking (Du et al., 2024)), bert2BERT allows us to increase both the width and depth dimensions of Transformer models (Vaswani et al., 2017), offering greater

---

[2]Note that the choice of $T_i^{\text{scratch}}$ is arbitrary depending on available resources, though most researchers follow the Chinchilla law (Hoffmann et al., 2022) as a principled approach to determine the compute-optimal number of training tokens. Our experimental setup also adopts the Chinchilla law (Section 4.2).

flexibility[3]. Specifically, for width expansion, bert2BERT increases the hidden dimensions by duplicating the weights of linear layers; for depth expansion, it stacks additional layers by duplicating pre-trained ones[4]. This flexibility to expand along both dimensions is particularly valuable as we repeatedly apply model expansion in progressive training.

## 4 Experiment

To evaluate the effectiveness of progressive training, we pre-train and compare two model families: one with independently training each model from scratch (`Independent`) and the other with progressive training (`Progressive`). Specifically, we demonstrate that progressive training improves computational efficiency over independent training, and the performance of the two model families is comparable.

### 4.1 Experimental Setup

We used FineWeb-Edu (Penedo et al., 2024) as the training data for pre-training and adopted the GPT-2 (Radford et al., 2019) tokenizer for tokenization. While progressive training can accommodate any parameter size increase, in this work, we adopted a configuration where the parameter count doubles at each stage. Specifically, we construct a model family with sizes $[X_1 = 1B, X_2 = 2B, X_3 = 4B, X_4 = 8B]$[5]. All models follow the Llama architecture (Touvron et al., 2023) with a maximum input sequence of 1024 tokens. Each model was trained using a cosine learning rate scheduler, with a maximum learning rate of $3.0 \times 10^{-4}$.

To evaluate each pre-trained model in the family, we measured perplexity on the validation data (Valid) from FineWeb-Edu and WikiText (Merity et al., 2017). To more comprehensively assess the performance of pre-trained models, we also evaluated zero-shot performance across multiple downstream tasks. Specifically, we include tasks including language modeling (LAMBADA (Paperno et al., 2016)), commonsense reasoning (WinoGrande (Sakaguchi et al., 2021), PIQA (Bisk et al., 2020), HellaSwag (Zellers et al., 2019)), and question answering (ARC-e, ARC-c (Clark et al., 2018), OBQA (Mihaylov et al., 2018)).

### 4.2 Training Data Size and Computational Cost

We prepare the following two data sizes, (i) Chinchilla law (Hoffmann et al., 2022) and (ii) 2x Chinchilla law, to determine the number of training tokens $T_i^{\text{scratch}}$ for each model in a model family. Chinchilla law provides the optimal FLOPs required to achieve a specific loss. The guideline is to use 20 tokens per model parameter for training (Hoffmann et al., 2022). 2x Chinchilla law is used to simulate the case such that the amount of training data greatly exceeds the optimal values indicated by the Chinchilla law. In fact, exceeding the Chinchilla law has become a standard practice in recent LLM literature (Sardana et al., 2024).

In the progressive training approach, we determine the number of training tokens $T_i^{\text{prog}}$ for each model to satisfy the constraint defined in Equation (4), ensuring it matches the total computational cost equals that of training the largest model from scratch. Under the 2x Chinchilla law setting, this constraint yields the following token allocations: $T_1^{\text{prog}} = 40B, T_2^{\text{prog}} = 60B, T_3^{\text{prog}} = 120B, T_4^{\text{prog}} = 240B$[6].

One of the key advantages of progressive training is its ability to obtain models of multiple sizes efficiently. As defined in Section 2, the computational cost is calculated as FLOPs =

---

[3]Although recent methods such as LEMON (Wang et al., 2024b) have emerged, LEMON builds upon the foundation of bert2BERT with only minor modifications to its core mechanisms. We choose bert2BERT for its simplicity and ease of implementation.

[4]The original bert2BERT paper (Chen et al., 2022) introduces two expansion variants: AKI and FPI. In this work, we exclusively adopt the AKI

[5]The specific width and depth settings for each model are detailed in Appendix A.

[6]For the Chinchilla law setting, $T_i^{\text{prog}}$ is shown in Appendix B.

| | | Perplexity ↓ | | Accuracy ↑ | | | | | | |
|---|---|---|---|---|---|---|---|---|---|---|
| | | Valid | Wikitext | LAMBADA | ARC-e | ARC-c | Winogrande | PIQA | OBQA | HellaSwag |
| 1B | Independent | 13.14 | 22.81 | 39.3 | 59.3 | 31.7 | 54.0 | 70.5 | 35.6 | 46.9 |
| 2B | Independent | 11.30 | **18.57** | **45.5** | **65.2** | **37.7** | 55.3 | 72.1 | 38.8 | 54.3 |
| | Progressive | **11.29** | 18.74 | 45.1 | 63.6 | 36.6 | **57.4** | **72.7** | **39.2** | **54.7** |
| 4B | Independent | 9.91 | 15.46 | 49.5 | 68.8 | **41.1** | 57.2 | 75.1 | **43.8** | 60.5 |
| | Progressive | **9.87** | **15.12** | **51.0** | **69.4** | 40.0 | **58.6** | 75.1 | 40.6 | **61.2** |
| 8B | Independent | 8.65 | 12.24 | 53.9 | 71.8 | 43.1 | **62.4** | 76.2 | 42.6 | 65.8 |
| | Progressive | **8.61** | **11.98** | **55.4** | **73.5** | **45.1** | 62.1 | **76.6** | **45.6** | **67.5** |

Table 1: Evaluation results of pre-trained models under the Chinchilla law setting. For each model size, we compare the performance of models trained independently from scratch (`Independent`) versus those built using progressive training (`Progressive`).

| | | Perplexity ↓ | | Accuracy ↑ | | | | | | |
|---|---|---|---|---|---|---|---|---|---|---|
| | | Valid | Wikitext | LAMBADA | ARC-e | ARC-c | Winogrande | PIQA | OBQA | HellaSwag |
| 1B | Independent | 12.04 | 20.56 | 42.0 | 62.3 | 34.6 | 55.5 | 72.0 | 35.0 | 51.6 |
| 2B | Independent | 10.44 | 17.11 | 48.5 | 66.9 | **38.4** | 56.6 | **75.3** | **41.6** | 58.0 |
| | Progressive | **10.43** | **16.94** | **50.3** | **70.0** | 38.1 | **58.5** | 73.8 | 41.4 | **59.0** |
| 4B | Independent | 9.18 | 13.99 | 51.7 | 71.7 | 43.2 | 59.2 | **76.7** | 40.8 | 63.3 |
| | Progressive | **9.00** | **13.63** | **51.9** | **72.6** | **43.9** | **61.0** | 76.1 | **43.0** | **65.6** |
| 8B | Independent | 7.85 | 10.23 | 55.2 | 74.5 | 47.4 | 62.4 | 77.0 | **46.4** | 67.9 |
| | Progressive | **7.74** | 10.21 | **57.8** | **75.3** | 46.7 | **64.6** | 77.3 | 44.6 | **70.2** |
| | +Fixed Data | 7.73 | 10.22 | 56.2 | 74.8 | **47.8** | 63.1 | **77.8** | 45.2 | **70.4** |

Table 2: Evaluation results of pre-trained models under the 2x Chinchilla law setting. Performance is compared between models trained independently from scratch (`Independent`) and those built using progressive training (`Progressive`). `Progressive+Fixed Data` indicates training on a fixed dataset of 320B tokens.

$6XT$, where $X$ denotes the number of model parameters and $T$ is the number of training tokens. For instance, independently training each model in the family $[1B, 2B, 4B, 8B]$ under the 2x Chinchilla law would require FLOPs of $[0.24Z, 0.96Z, 3.84Z, 15.4Z]$ respectively. Thus, constructing the entire model family would require 20.4ZFLOPs in total.

In contrast, with progressive training, the total computational cost required to construct the entire model family is equivalent to the computational cost required to train the largest-size, i.e., 8B, model. Therefore, the computational cost is 15.4ZFLOPs. Compared to training each model independently, progressive training can reduce computational cost by approximately 25%. This demonstrates that progressive training improves the computational efficiency of constructing a model family.

### 4.3 Results

**Model Performance.** Tables 1 and 2 present the evaluation results of pre-trained models in the Chinchilla law and 2x Chinchilla law settings, respectively. Our progressive training approach (`Progressive`) achieves performance comparable to or better than models that are independently trained from scratch (`Independent`) across different parameter sizes in both settings. Notably, for the largest model (8B), `Progressive` shows improvements in perplexity and most downstream tasks.

**Computational Cost.** In Figure 2 (*Left*), for a given computational budget, i.e., ZFLOPs, `Progressive` consistently outperforms `Independent`. In addition, `Progressive` reaches the same training loss as `Independent` while reducing the total FLOPS by 26%.

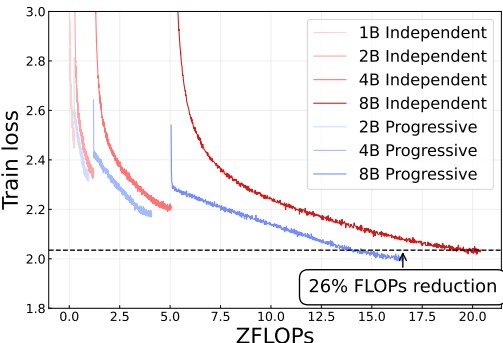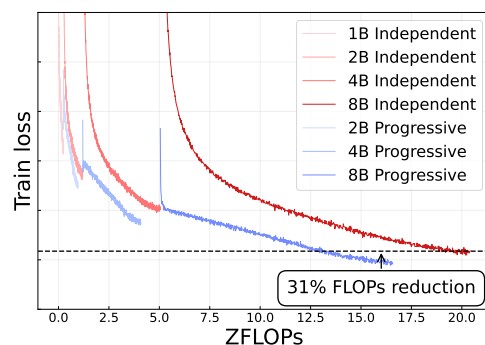

Figure 2: Train loss curves comparing models trained by `Independent` versus `Progressive` approach with the 2x Chinchilla law setting. (*Left*): Models trained with fixed maximum learning rate, achieving 26% FLOPs reduction. (*Right*): Models trained with maximum learning rate adjustment, from $1.5 \times 10^{-3}$ (1B) to $3.0 \times 10^{-4}$ (8B), achieving 31% FLOPs reduction.

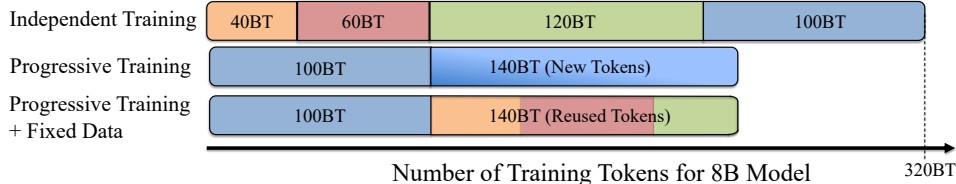

Figure 3: Comparison of token allocation for the 8B model training. (*Top*): `Independent` uses a standard approach with 320B tokens. (*Middle*): `Progressive` uses 240B tokens (100B + new 140B tokens). (*Bottom*): `Progressive+Fixed Data` maintains the same 240B total tokens as `Progressive` by reusing 140B tokens from previous stages.

**Does Progressive Training Exploit More Unique Data? No.** For each model size $X_i (i > 1)$ in our model family, we use fewer tokens in progressive training than independent training ($T_i^{\text{prog}} < T_i^{\text{scratch}}$). However, when considering the entire model family construction process, progressive training consumes more total tokens than training only the largest model from scratch. As detailed in Section 4.2, the total number of tokens processed in progressive training is $\sum_{i=1}^{4} T_i^{\text{prog}} = 460\text{B}$ tokens, which exceeds the $T_4^{\text{scratch}} = 320\text{B}$ tokens used when training the 8B model independently, even though the total computational cost is the same (Figure 3, (*Top*) and (*Middle*)). Thus, it is possible that progressive training benefits unfairly from exposure to a larger volume of unique training data. To isolate this factor, we introduced a controlled setting denoted as `Progressive+Fixed Data`, in which total amount of unique data is capped at 320B tokens – the same amount used when independently training the 8B model. Once the 320B tokens were consumed during the training of the 8B model, the remaining $460 - 320 = 140\text{B}$ tokens are reused, i.e., the second epoch begins, as illustrated in Figure 3 (*Bottom*).

The result is shown in Table 2, in which 8B `Progressive+Fixed Data` achieves comparable performance. This confirms that the observed improvements from progressive training are not attributable to access to a greater quantity of unique data.

### 4.4 Effectiveness of Maximum Learning Rate Adjustment

In LLM pre-training, selecting an appropriate learning rate requires a careful balance between optimization efficiency and training stability. Higher learning rates can accelerate convergence and potentially lead to better final model performance (Takase et al., 2023), but they must be tuned carefully, especially with respect to model size. As models scale up,

| | | Perplexity ↓ | | Accuracy ↑ | | | | | | |
|---|---|---|---|---|---|---|---|---|---|---|
| | | Valid | Wikitext | LAMBADA | ARC-e | ARC-c | Winogrande | PIQA | OBQA | HellaSwag |
| 1B | Independent | 12.04 | 20.56 | 42.0 | 62.3 | 34.6 | 55.5 | 72.0 | 35.0 | 51.6 |
| 2B | Independent | 10.44 | 17.11 | 48.5 | 66.9 | 38.4 | 56.6 | 75.3 | 41.6 | 58.0 |
| | Progressive | **10.07** | **16.17** | **52.6** | **71.8** | **42.2** | **61.9** | **75.0** | 40.8 | **62.5** |
| 4B | Independent | 9.18 | 13.99 | 51.7 | 71.7 | 43.2 | 59.2 | 76.7 | 40.8 | 63.3 |
| | Progressive | **8.72** | **13.13** | **55.7** | **74.5** | **47.8** | **65.2** | **77.4** | **45.8** | **68.1** |
| 8B | Independent | 7.85 | 10.23 | 55.2 | 74.5 | 47.4 | 62.4 | 77.0 | 46.4 | 67.9 |
| | Progressive | **7.64** | 10.10 | **59.0** | **76.7** | 48.3 | **65.8** | **78.3** | 46.6 | 71.0 |
| | +Fixed Data | **7.64** | **9.77** | 58.0 | 76.1 | **50.6** | 64.7 | 78.2 | **47.2** | **71.2** |

Table 3: Evaluation results of pre-trained models with the maximum learning rate adjustment strategy described in Section 4.4. `Progressive` models consistently outperform `Independent` models. `+Fixed Data` indicates training on a fixed dataset of 320B tokens.

they become increasingly sensitive to the learning rate, typically requiring smaller learning rates to preserve training stability (Touvron et al., 2023; Wortsman et al., 2024). For example, in our preliminary experiments, we observed that a learning rate of $1.5 \times 10^{-3}$ enabled effective and stable training for a 1B model. However, applying the same learning rate to an 8B model resulted in the loss value spikes that ultimately led to training collapse[7].

Progressive training method is characterized by starting training from smaller models. Therefore, it is possible to employ high learning rates to enhance performance during small-model training while lowering the learning rate for larger models to stabilize training. In this experiment, we apply this maximum learning rate adjustment under the 2x Chinchilla law setting, linearly decreasing the maximum learning rate from $1.5 \times 10^{-3}$ for the 1B model to $3.0 \times 10^{-4}$ for the 8B model, in accordance with the increase in model size, while maintaining a cosine decay schedule during training for all models[8].

Table 3 presents the results of our maximum learning rate adjustment strategy. `Progressive` consistently outperforms the `Independent` counterparts, with more substantial improvements than those observed with no learning rate adjustments (Table 2). As illustrated in Figure 2 (*Right*), our maximum learning rate adjustment facilitated more efficient convergence during training, achieving 31% FLOPs reduction compared to 26% reduction observed without learning rate adjustment. These results demonstrate the effectiveness of our progressive training approach and show that appropriate learning rate adjustments can further enhance performance[9].

# 5 The Consistency across Model Family

In this section, we investigate the consistency of model behaviors across different sizes in our progressively trained model family. We first discuss the advantages of maintaining consistent behaviors across the model family (Section 5.1). We then present empirical analyses of consistency through probability distribution and speculative decoding. Our findings show that models trained using the progressive training approach exhibit higher consistency compared to independently trained models.

## 5.1 The Potential Effectiveness of Consistency across Model Family

Ensuring consistent behavior across a model family offers several practical benefits. First, it simplifies deployment flexibility by ensuring that switching between different parame-

---

[7]A visualization of this training instability with the 8B model is provided in the Appendix C.

[8]The learning rates for each model size are described in Appendix D.

[9]Moreover, as shown in Appendix E the benefits of our approach persist under post-training setups (SFT+DPO), confirming its robustness for practical applications.

| Model Pair | Training Approach | $D_{\text{KL}}(P_{X_i} \| P_{X_{i+1}})$ |
|---|---|---|
| | Independent | 0.2821 |
| 1B→2B | Progressive | **0.2162** |
| | Progressive+LR adjustment | 0.2538 |
| | Independent | 0.3087 |
| 2B→4B | Progressive | **0.2265** |
| | Progressive+LR adjustment | 0.2542 |
| | Independent | 0.4584 |
| 4B→8B | Progressive | **0.3378** |
| | Progressive+LR adjustment | 0.3518 |

Table 4: KL divergence between adjacent model sizes in the different model families. Lower values indicate greater consistency. Results are shown for models trained from scratch (`Independent`), models built through progressive training (`Progressive`), and models built with maximum learning rate adjustment (`Progressive+LR adjustment`).

ter sizes does not lead to large, sudden shifts in model outputs or user experience. This is particularly valuable when developers need to adapt a model size to varying computational budgets or runtime constraints, as they can easily scale up or down (e.g., from a smaller to a larger model) without retraining users or extensively revalidating system performance (Srivastava et al., 2020; Echterhoff et al., 2024).

Second, such consistency enables more efficient incremental improvements or patches across the entire model family. For instance, if developers collect preference data or build a reward model based on the outputs of one model (e.g., by annotating the specific texts that this member tends to generate), they typically face a distribution mismatch when applying those artifacts to other models. In a consistent model family, however, differences in generation patterns are small enough examples or learned preference remain effective for other models (Guo et al., 2024; Zhou et al., 2024a; Tajwar et al., 2024). Likewise, implementations such as safety or content filters calibrated for one model will likely transfer to others with only minimal adjustments (Inan et al., 2023).

Third, consistency across models of different sizes benefits speculative decoding (Leviathan et al., 2023), a technique that accelerates inference by allowing a smaller model to generate "draft" outputs, which the larger model then either accepts or refines. When the smaller and larger models produce similar probability distributions, the larger model is more likely to accept the drafts, reducing the frequency of rejections and subsequent regenerations (Zhou et al., 2024b). Consequently, consistent behavior across a range of model sizes further contributes to more efficient and user-friendly deployments in real-world applications.

## 5.2 Probability Distribution Consistency across Model Family

The consistency of the underlying probability distributions across the model family offers deeper insights into how knowledge and behaviors are propagated. Such consistency indicates that models rely on similar internal mechanisms when generating text.

**Experimental Setup.** To evaluate the probability distribution consistency across our model family, we measured the KL divergence between adjacent model sizes. KL divergence quantifies how one probability distribution differs from another reference distribution, with lower values indicating greater similarity. For this analysis, we used 10,000 examples from the FineWeb-Edu validation dataset. We calculated the KL divergence $D_{\text{KL}}(P_{X_i} \| P_{X_{i+1}})$ between adjacent models by examining their next-token prediction distributions, where $P_{X_i}$ represents the probability distribution over the vocabulary given by model of size $X_i$[10]. We conducted this analysis for three different model families: (i) Models trained independently from scratch, (ii) Models built through progressive training, (iii) Models built through progressive training with maximum learning rate adjustment.

---

[10]The detailed calculation method for KL divergence is described in Appendix F.

| Draft Model (4B) | Generator Model (8B) | $D_{\mathrm{KL}}(P_{\mathrm{draft}} \parallel P_{\mathrm{generator}})$ | Acceptance Rate (%) | Generation Time (s) |
|---|---|---|---|---|
| Progressive | Progressive | **0.3378** | **93.20** | **7.02** |
| Independent | Progressive | 0.4281 | 87.20 | 8.24 |

Table 5: Speculative decoding performance with different draft-generator configurations. Progressive draft models demonstrate lower KL divergence with the generator, resulting in higher acceptance rates and faster generation times.

**Results.** Table 4 presents the KL divergence between adjacent model sizes for our three different model families. We found that, across all model pairs, models trained via progressive training demonstrate substantially lower KL divergence compared to independently trained models, indicating higher consistency in their probability distributions. This trend is consistent across all adjacent model pairs (1B→2B, 2B→4B, 4B→8B).

This finding provides evidence that our progressive training produces a more coherent model family with consistent internal behavior. The lower KL divergence suggests that progressively trained models share similar probability distributions and decision-making processes, offering practical advantages for techniques like speculative decoding, where alignment between smaller and larger models is crucial.

## 5.3 Validating Consistency Benefits through Speculative Decoding

We empirically assess whether this consistency translates into practical benefits for speculative decoding, as discussed in Section 5.1.

**Experimental Setup.** In speculative decoding, a smaller "draft" model proposes multiple tokens that a large "generator" model subsequently accepts or rejects. For our experiments, we fixed the 8B model by Progressive training as the generator and paired it with 4B draft models trained using either Progressive or Independent training. Since the effectiveness of speculative decoding depends on how well the draft model's predictions align with the generator's distribution, this setup validates whether the distributional consistency yields practical benefits. Each draft model was configured to propose 8 speculative tokens. To evaluate speculative decoding performance, we measured three key metrics on 1,000 prompts sampled from the FineWeb-Edu validation dataset. First, we computed the KL divergence $D_{\mathrm{KL}}(P_{X_{\mathrm{draft}}} \parallel P_{X_{\mathrm{generator}}})$ between the draft and generator models' output distributions. Second, we tracked the acceptance rate, which is the percentage of draft tokens accepted by the generator model. This metric directly reflects the efficiency of speculative decoding. Finally, we recorded the average generation time across all prompts to quantify the practical speedup.

**Results.** Table 5 presents the speculative decoding performance with different draft-generator configurations. The results demonstrate the practical advantages of employing a draft model constructed through Progressive Training. The pairing of a 4B Progressive draft model with an 8B Progressive generator model (Prog-Prog) exhibited superior performance, underpinned by greater consistency between the two models. This enhanced consistency is quantified by a lower KL divergence for the Prog-Prog pair, compared to using an independently trained model as a drafter (Inde-Prog). This improved alignment in output distributions directly translated to more effective speculative decoding: the Prog-Prog configuration achieved a higher acceptance rate of 93.20%, as opposed to 87.2% for the Inde-Prog setup. Consequently, this led to a reduction in inference time, with the Prog-Prog pair completing generation approximately 14.8% faster. These findings experimentally demonstrate that the consistency fostered by Progressive Training offers practical benefits. The lower KL divergence between progressively trained models leads to more effective draft proposals in speculative decoding, thereby improving acceptance rates and reducing inference latency.

## 6 Related Work

**Model Family.** Offering models in multiple sizes has become a standard practice in language model development to address diverse computational requirements. Major research labs have released model families with multiple parameter sizes: Llama (Touvron et al., 2023; AI@Meta, 2024), Qwen (Yang et al., 2024a), Gemma (GemmaTeam, 2024), and others. Scaling laws (Kaplan et al., 2020; Hoffmann et al., 2022) provide the theoretical foundation for these families, establishing relationships between model size, data, compute, and performance. Recent research highlights the complementary roles of models of different sizes (Wang et al., 2024a): while large models excel in zero/few-shot generalization, smaller models offer advantages for latency-sensitive applications, edge deployments, domain-specific tasks, and privacy-sensitive contexts. This functional differentiation underscores the importance of efficient methods for constructing a coherent model family that maintains consistent capabilities across different parameter sizes. However, the conventional approach of training each model size independently incurs substantial computational costs that scale additively with the number of models in the family. Our work addresses this inefficiency by proposing progressive training that significantly reduces the total computation required to construct a complete and coherent model family.

**Model Expansion.** Model expansion has emerged as an approach to reduce computational costs when training large-scale models. These methods leverage parameters of smaller, pre-trained models to initialize larger ones, accelerating training. The bert2BERT (Chen et al., 2022) adapted function-preserving methods to Transformers, reducing pre-training costs by approximately 45% for BERT models. LEMON (Wang et al., 2024b) refined this approach with an optimized learning rate scheduler for larger models. For scaling LLMs, Du et al. (2024) identified depthwise stacking as particularly effective, achieving 54.6% speedup for 7B-parameter models. Alternative approaches combine knowledge distillation with model expansion. Qin et al. (2022) introduced Knowledge Inheritance (KI), using knowledge distillation during pre-training to achieve approximately 27% computational cost reduction. However, these methods typically focus on the efficient training of a single, final model rather than the training of an entire model family. For example, many methods demonstrate the effectiveness of the expansion in a single-shot manner, i.e., applying model expansion only once. In addition, these methods do not take the performance of intermediate model(s) into account. In contrast, we propose progressive training and demonstrate that the off-the-shelf model expansion methods can be used to efficiently train a model family. The progressive training can incorporate emerging novel model expansion techniques, further enhancing efficiency.

## 7 Conclusion

In this paper, we proposed an efficient approach for constructing a model family via progressive training, wherein smaller models are incrementally expanded to larger sizes. Through comprehensive experiments on a model family ranging from 1B to 8B parameters, we demonstrated that our method significantly reduces the total computational cost required to construct a complete model family by approximately 25% compared to training each model independently. Our method, particularly when combined with maximum learning rate adjustments tailored to each model size, not only matches but exceeds the performance of independently trained models. Furthermore, models built through our progressive approach demonstrate greater consistency across different sizes, as evidenced by lower KL divergence between probability distributions.

This work offers researchers and practitioners with a computationally efficient approach to constructing a high-performing, coherent model family, offering particular value in resource-constrained environments where training multiple models at different scales is desirable.

## Acknowledgements

This work was supported by the JST Moonshot R&D Grant Number JPMJMS2011-35 (fundamental research).

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

| Configuration | 1B | 2B | 4B | 8B |
|---|---|---|---|---|
| Hidden dimension | 2048 | 2560 | 3200 | 4096 |
| FFN dimension | 7168 | 8960 | 11200 | 14336 |
| Layers | 18 | 22 | 27 | 33 |
| Heads | 16 | 20 | 25 | 32 |
| Batch size | 960 | 1920 | 3840 | 7680 |
| The number of updates | 40700 | 40700 | 40700 | 40700 |
| learning rate warmup fraction | 0.01 | 0.01 | 0.01 | 0.01 |
| Learning rate decay style | cosine | cosine | cosine | cosine |
| Adam $\beta_1$ | 0.9 | 0.9 | 0.9 | 0.9 |
| Adam $\beta_2$ | 0.95 | 0.95 | 0.95 | 0.95 |
| Gradient clipping | 1.0 | 1.0 | 1.0 | 1.0 |
| Weight decay | 0.1 | 0.1 | 0.1 | 0.1 |
| Precision | `bfloat16` | `bfloat16` | `bfloat16` | `bfloat16` |

Table 6: Experimental configuration for each model size

An Yang, Baosong Yang, Binyuan Hui, Bo Zheng, Bowen Yu, Chang Zhou, Chengpeng Li, Chengyuan Li, Dayiheng Liu, Fei Huang, Guanting Dong, Haoran Wei, Huan Lin, Jialong Tang, Jialin Wang, Jian Yang, Jianhong Tu, Jianwei Zhang, Jianxin Ma, Jianxin Yang, Jin Xu, Jingren Zhou, Jinze Bai, Jinzheng He, Junyang Lin, Kai Dang, Keming Lu, Keqin Chen, Kexin Yang, Mei Li, Mingfeng Xue, Na Ni, Pei Zhang, Peng Wang, Ru Peng, Rui Men, Ruize Gao, Runji Lin, Shijie Wang, Shuai Bai, Sinan Tan, Tianhang Zhu, Tianhao Li, Tianyu Liu, Wenbin Ge, Xiaodong Deng, Xiaohuan Zhou, Xingzhang Ren, Xinyu Zhang, Xipin Wei, Xuancheng Ren, Xuejing Liu, Yang Fan, Yang Yao, Yichang Zhang, Yu Wan, Yunfei Chu, Yuqiong Liu, Zeyu Cui, Zhenru Zhang, Zhifang Guo, and Zhihao Fan. Qwen2 technical report, 2024a. URL https://arxiv.org/abs/2407.10671.

An Yang, Baosong Yang, Beichen Zhang, Binyuan Hui, Bo Zheng, Bowen Yu, Chengyuan Li, Dayiheng Liu, Fei Huang, Haoran Wei, et al. Qwen2. 5 technical report. *arXiv preprint arXiv:2412.15115*, 2024b.

Rowan Zellers, Ari Holtzman, Yonatan Bisk, Ali Farhadi, and Yejin Choi. HellaSwag: Can a machine really finish your sentence? In Anna Korhonen, David Traum, and Lluís Màrquez (eds.), *Proceedings of the 57th Annual Meeting of the Association for Computational Linguistics*, pp. 4791–4800, Florence, Italy, July 2019. Association for Computational Linguistics. doi: 10.18653/v1/P19-1472. URL https://aclanthology.org/P19-1472.

Lianmin Zheng, Wei-Lin Chiang, Ying Sheng, Siyuan Zhuang, Zhanghao Wu, Yonghao Zhuang, Zi Lin, Zhuohan Li, Dacheng Li, Eric Xing, Hao Zhang, Joseph E. Gonzalez, and Ion Stoica. Judging LLM-as-a-judge with MT-bench and chatbot arena. In *Thirty-seventh Conference on Neural Information Processing Systems Datasets and Benchmarks Track*, 2023. URL https://openreview.net/forum?id=uccHPGDlao.

Wenxuan Zhou, Ravi Agrawal, Shujian Zhang, Sathish Reddy Indurthi, Sanqiang Zhao, Kaiqiang Song, Silei Xu, and Chenguang Zhu. WPO: Enhancing RLHF with weighted preference optimization. In *Proceedings of the 2024 Conference on Empirical Methods in Natural Language Processing*, November 2024a. doi: 10.18653/v1/2024.emnlp-main.475. URL https://aclanthology.org/2024.emnlp-main.475/.

Yongchao Zhou, Kaifeng Lyu, Ankit Singh Rawat, Aditya Krishna Menon, Afshin Rostamizadeh, Sanjiv Kumar, Jean-François Kagy, and Rishabh Agarwal. Distillspec: Improving speculative decoding via knowledge distillation. In *The Twelfth International Conference on Learning Representations*, 2024b. URL https://openreview.net/forum?id=rsY6J3ZaTF.

## A   Details of Experimental Configurations

In our experiments (Section 4), we constructed models with 1B, 2B, 4B, and 8B parameters based on the Llama architecture. Table 6 shows the specific experimental configurations for

| Model Size | $T_i^{\text{scratch}}$ (Independent) | $T_i^{\text{prog}}$ (Progressive) |
|:---:|:---:|:---:|
| 1B | 20B | 20B |
| 2B | 40B | 30B |
| 4B | 80B | 60B |
| 8B | 160B | 120B |

Table 7: Training token allocation for `Independent` and `Progressive` the Chinchilla law setting.

| Model Size | $T_i^{\text{scratch}}$ (Independent) | $T_i^{\text{prog}}$ (Progressive) |
|:---:|:---:|:---:|
| 1B | 40B | 40B |
| 2B | 80B | 60B |
| 4B | 160B | 120B |
| 8B | 320B | 240B |

Table 8: Training token allocation for `Independent` and `Progressive` the 2x Chinchilla law setting.

each model size. As the model expands, we gradually increase the hidden dimension, FFN dimension, number of layers, and number of heads.

## B  Data Size for (2x) Chinchilla Law Setting

As described in Section 4.2, we determined the number of training tokens for each model size according to Hoffmann et al. (2022). We report token allocations for each model in Table 7 (Chinchilla law setting) and Table 8 (2x Chinchilla law setting).

## C  8B Model Loss Visualization

Figure 4 illustrates the training loss curves for 8B models trained from scratch with different learning rates: $3.0 \times 10^{-4}$ and $1.5 \times 10^{-3}$. While the lower learning rate of $3.0 \times 10^{-4}$ results in stable training, the higher learning rate of $1.5 \times 10^{-3}$ causes significant loss spikes during training.

## D  Specific Configurations in the Learning Rate Adjustment Strategy

In our learning rate adjustment strategy (Section 4.4), we decrease the maximum learning rate as the model size increases. We apply maximum learning rates of $1.5 \times 10^{-3}$, $1.1 \times 10^{-3}$, $7.0 \times 10^{-4}$, and $3.0 \times 10^{-4}$ for the 1B, 2B, 4B, and 8B models, respectively. This gradual reduction accommodates the increasing sensitivity of larger models to high learning rates.

## E  Experiment: Model Family Post-Training

This experiment verifies whether the trends observed in pre-training experiments in Section 4 are also seen in post-training. We examine whether models built using progressive training with fixed learning rates can achieve comparable performance to models trained from scratch. In addition, we confirm that performance improvements observed with maximum learning rate adjustments in pre-training also extend to post-training.

### E.1  Experimental Setup

We evaluate the effectiveness of progressive training by conducting post-training on the model family constructed in Section 4. We adopted the settings from Meng et al. (2024) for

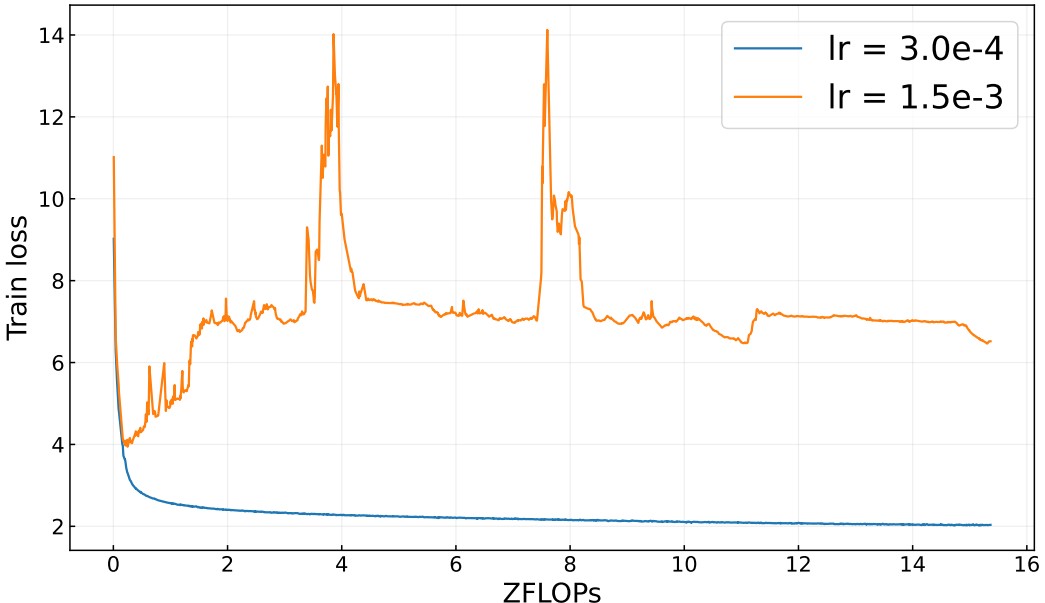

Figure 4: Training loss curves comparing 8B models trained from scratch with different learning rates. The model trained with a learning rate of $3.0 \times 10^{-4}$ (blue line) shows stable training, while the model trained with a higher learning rate of $1.5 \times 10^{-3}$ (orange line) exhibits severe loss spikes indicating training instability.

| Size | Training Approach | Chinchilla MT-Bench ↑ | 2x Chinchilla MT-Bench ↑ | Adjusted LR 2x Chinchilla MT-Bench ↑ |
|---|---|---|---|---|
| 2B | Independent | 1.61 | 2.04 | 2.04 |
|    | Progressive | **2.03** | **2.46** | **3.22** |
| 4B | Independent | 2.78 | **3.02** | 3.02 |
|    | Progressive | **2.87** | 3.01 | **3.63** |
| 8B | Independent | **3.42** | **3.45** | 3.45 |
|    | Progressive | 3.28 | 3.21 | **3.96** |
|    | Progressive+Fixed Data | 3.32 | 3.41 | **3.98** |

Table 9: MT-Bench evaluation results after post-training (SFT+DPO) for models trained with different approaches. We compare models trained from scratch (`Independent`) versus those built using our progressive training approach (`Progressive`) under both fixed maximum learning rate and learning rate adjustment (Adjusted LR) strategies. For the fixed learning rate, results are shown for both the Chinchilla law and 2x Chinchilla law settings. For the models with learning rate adjustments, the maximum learning rates are decreased from $1.5 \times 10^{-3}$ for 1B to $3.0 \times 10^{-4}$ for 8B as the model size increases.

post-training. Specifically, we performed Supervised Fine-Tuning (SFT) using the UltraChat-200k dataset (Ding et al., 2023), followed by DPO (Rafailov et al., 2023) using the Ultrachat Feedback dataset (Cui et al., 2024).

The maximum learning rate was set to $3.0 \times 10^{-5}$ for SFT and $5.0 \times 10^{-7}$ for DPO, standardized across all models. We evaluated the response quality of each model using MT-Bench (Zheng et al., 2023) using gpt-4-0613 as an evaluator. For comparison, we prepared models (2B, 4B, 8B) independently trained from scratch and conducted the same post-training.

### E.2 Results

Table 9 presents the MT-Bench evaluation results after post-training across different training approaches. With fixed learning rates, models trained with progressive training perform comparably to models trained from scratch, with some variation across model sizes.

When applying our maximum learning rate adjustment, progressive training consistently outperforms from-scratch training across all model sizes, with substantial improvements observed. The models with maximum learning rate adjustment significantly outperform those with fixed maximum learning rates for progressive training.

These results indicate that our progressive training approach, particularly when combined with maximum learning rate adjustment, can deliver both computational efficiency and enhanced model performance after post-training.

## F  KL Divergence Computation

To simplify the computation, at each token position, we computed $D_{\mathrm{KL}}(P_{X_i} \parallel P_{X_{i+1}}) = \sum_{v \in V} P_{X_i}(v) \cdot (\log P_{X_i}(v) - \log P_{X_{i+1}}(v))$, where $V$ is the vocabulary and $P_{X_i}(v)$ is the probability assigned to vocabulary token $v$ by model $X_i$. We first averaged these KL divergence values across all token positions within each example, then computed the final KL divergence by averaging across all 10,000 examples from the FineWeb-Edu validation dataset, as described in Section 5.2.

