# OpenReview forum: "Efficient Construction of Model Family through Progressive Training Using Model Expansion"
_colmweb.org/COLM/2025/Conference — COLM 2025_

### Official Review · Reviewer_Az33 · 2025-05-11

**Rating:** 6
**Confidence:** 3
**Ethics Flag:** 1

**Summary:**

This paper aims to train a family of LLMs efficiently. Specifically, the authors propose to progressively train the model family from small to large ones. When training a larger model, bert2BERT method is utilized to initialize the larger model from smaller ones. Experiments show good results, with comparable performance while reducing computation overhead.

**Reasons To Accept:**

1. This paper tackles an important and interesting problem. Training a family of models is important.
2. The ideas are well validated by good experimental results in convincing settings.

**Reasons To Reject:**

1. The idea is simply to apply the bert2BERT method to LLMs for model expansion. Although this new experimental setting is good and interesting, the novelty is relatively limited.
2. The intuition and insight are not well illustrated. Why can progressive training lead to better results? How does this intuition guide the design choices of each component? This can be made more clear.
3. Missing reference. There are many model expansion works in the literature. This paper cites only very few of them. The missing reference can be supplemented. For example, Knowledge Inheritance for Pre-trained Language Models. NAACL 2022.

---

> ### Author Response · Authors · 2025-06-02
>
> Thank you for your feedback. We would like to address the weaknesses as follows:
>
> ## Weakness 1.
> > The idea is simply to apply the bert2BERT method to LLMs for model expansion. Although this new experimental setting is good and interesting, the novelty is relatively limited.
>
> Our novelty lies in (1) framing a practical but previously overlooked application—model family construction—as an evaluable problem and (2) empirically demonstrating that tailoring model expansion techniques for this purpose is highly effective.
>
> ## Weakness 2.
> > The intuition and insight are not well illustrated. Why can progressive training lead to better results? How does this intuition guide the design choices of each component? This can be made more clear.
>
> The core intuition is that model expansion provides superior initialization by transferring knowledge from smaller models, placing larger models in better regions of the loss landscape compared to random initialization [1].
> Our primary contribution is computational efficiency. The performance improvements, while encouraging, represent a bonus rather than the main objective. Better initialization leading to better performance is well-established in neural network optimization [2]. Understanding the precise mechanisms in our setting presents interesting future work.
>
> ## Weakness 3.
> > Missing reference. There are many model expansion works in the literature. This paper cites only very few of them. The missing reference can be supplemented. For example, Knowledge Inheritance for Pre-trained Language Models. NAACL 2022.
>
> Our Related Work section currently cites representative and foundational works in model expansion that are directly relevant to our progressive training approach, providing sufficient coverage of the core techniques and recent developments in the field.
> Nevertheless, we recognize the value of providing a more comprehensive literature review and specifically appreciate you pointing out "Knowledge Inheritance for Pre-trained Language Models" (Qin et al., 2022). While Qin et al. (2022) explore the related and important area of leveraging smaller models to efficiently train larger ones (primarily via knowledge distillation ), our work on progressive training using model expansion offers distinct contributions. Our primary focus is the efficient construction of an entire model family by iteratively applying model expansion. This approach aims to reduce the total computational cost for the family significantly, ideally to that of training only the largest model, while also demonstrating benefits in performance and cross-model behavioral consistency.
>
> ### References.
> [1] Chen et al. NAACL 2022. bert2BERT: Towards Reusable Pretrained Language Models.
>
> [2] Qi et al. NAACL 2021. When and Why Are Pre-Trained Word Embeddings Useful for Neural Machine Translation?

---

> > ### Author Response · Authors · 2025-06-10
> >
> > With just a day remaining until the discussion period concludes, we wanted to express our appreciation for your review of our manuscript. We would appreciate it if you could let us know if you still have any concerns or if there are any remaining points that need further clarification.

---

### Official Review · Reviewer_oPhZ · 2025-05-12

**Rating:** 8
**Confidence:** 3
**Ethics Flag:** 1

**Summary:**

This paper proposes progressive model expansion. Compared with single‑shot bert2BERT or LEMON, the novelty is treating every intermediate checkpoint as a usable model and tuning the LR per stage. On a FineWeb‑Edu pre‑train + SFT + DPO pipeline, the 8 B progressive family matches or slightly beats scratch baselines while saving around 25 – 31 % compute and yielding smoother KL between sizes.

**Questions To Authors:**

1. Does the method remain stable at even larger models?
2. How would depth‑only or LEMON expansion change results?
3. What is the val‑ppl change on reused tokens in the Fixed‑Data ablation?

**Reasons To Accept:**

- Simple & reproducible: re‑uses public bert2BERT and standard optimisers.
- Thorough evaluation: 2 compute budgets, 8 downstream tasks, etc.
- Smoother speculative‑decoding pipeline thanks to lower intra‑family KL.
- Practical impact: ‑25 % compute at 8 B saves thousands of GPU‑hours.

**Reasons To Reject:**

- Using LLM up to 8 B -- unclear at ≥70 B.
- Incremental novelty: cascaded expansion is a straightforward extension.
- Compute vs. data: progressive run still sees 2× epochs (possible confound).
- No robustness or safety analysis: bias and toxicity are not reported.
- There might be more related work relevant to be mentioned (e.g., https://www.arxiv.org/abs/2502.07832, https://arxiv.org/abs/2402.11700).

---

> ### Author Response · Authors · 2025-06-02
>
> Thank you for your positive feedback. We would like to answer your questions as follows:
>
> ## Answer to *Does the method remain stable at even larger models?*
> Thank you for your question. If 'stable' refers to maintaining computational efficiency gains, we believe our progressive training method should indeed hold its efficiency for even larger models.
> Consider a general model family where each model doubles in size (e.g., [X,2X,4X,8X]). Our paper defines the computational cost (FLOPs) as $FLOPs = 6 \times X \times T$. Adopting the Chinchilla law of using 20 tokens per model parameter for training from scratch ($T_{i}^{\mathrm{scratch​}}=20 \times X_i​$), the costs for independent training would be [120$X^2$, 480$X^2$, 1920$X^2$, 7680$X^2$].
> With our progressive training approach, the total computational cost to build the entire family is equivalent to the cost of training the largest model (8X in this case) from scratch. So, the percentage reduction in computational cost is therefore ($2520X^2/10200X^2) \times 100 \approx 24.7$%.
> As long as similar scaling proportions are used for the family, this percentage saving should be largely preserved even as the absolute size X increases.
>
> ## Answer to *How would depth‑only or LEMON expansion change results?*
>
> Thank you for this insightful question. While our study demonstrates that bert2BERT's capability to expand both width and depth dimensions is sufficient for effective progressive training, exploring alternative model expansion methods is indeed valuable.
>
> **Experiment:**
>
>  To address this question, we conducted additional experiments using depth-only expansion (specifically, layer stacking as employed in recent works [1] ) on a smaller model family: 500M → 1B → 2B parameters under the Chinchilla setting.
>
> **Results:**
>
> |  Model    |   Method  | Validation  Perplexity ↓  |
> |-|-|-|
> | 500M       | Independent | 15.08         |
> | 1B      | Independent |    12.74      |
> | 1B      | Progressive | **12.47** |
> | 2B          | Independent | 10.90         |
> | 2B          | Progressive | **10.59** |
>
> The above table shows that depth-only stacking also enables progressive training to achieve comparable or better performance than independent training, while constructing both 500M and 1B models using only the computational budget required for training the 2B model independently. These results are consistent with our main findings using bert2BERT.
> Future work should explore how different model expansion methods perform within the progressive training framework to further optimize the approach.
>
> ## Answer to *What is the val‑ppl change on reused tokens in the Fixed‑Data ablation?*
>  The slight performance variations are expected due to different data ordering and sampling during training.
> This comparable performance is consistent with prior work showing that training with repeated data up to ~4 epochs maintains similar performance without significant degradation [2]. In our Progressive+Fixed Data setting, we reuse 140B tokens (less than 1.5 epochs of repetition), which falls well within the range where data repetition doesn't harm performance.
>
> ### References.
> [1] Du et al. NeurIPS 2024. Stacking Your Transformers: A Closer Look at Model Growth for Efficient LLM Pre-Training.
>
> [2] Muennighoff et al. NeurIPS 2023. Scaling Data-Constrained Language Models.

---

> > ### Author Response · Authors · 2025-06-10
> >
> > With just a day remaining in the discussion period, we wanted to express our gratitude for the time you invested in reviewing our manuscript and for your positive evaluation.

---

### Official Review · Reviewer_D8FB · 2025-05-12

**Rating:** 6
**Confidence:** 4
**Ethics Flag:** 1

**Summary:**

This paper proposed a new method for training a LLM series containing models of varying sizes. Instead of training models of different sizes independently (from scratch), the authors propose to only train the smallest model from scratch, while larger models are instantiated from the immediately smaller model using the model expansion method, bert2BERT (AKI). Empirical results show that this method, named progressive training, can reduce the total training FLOPs of the model series by roughly 25%, while getting models with more consistent behavior (output distributions).

**Questions To Authors:**

* Why does using repeated data outperform the model that uses unique data (shown in Table 2)?

* Can we perform progressive training in a more fine-grained manner? For instance, start from 1M parameters and progressively scale up to 8B?

* Is the expansion method compatible with muP [1], which has become a popular technique in LLM training?

* I think it is more common to have a space between the number before “ZFLOPs” in, for instance, “20.4ZFLOPs”

References:

[1] Yang et al. 2021. “Tensor Programs V: Tuning Large Neural Networks via Zero-Shot Hyperparameter Transfer”

**Reasons To Accept:**

* The paper is very well-written. The motivation is clear, the presentation is very easy to follow.

* The method is effective and reasonable. It also has great practical value and is interesting to many LLM practitioners.

* The method is validated on up to models up to 8B parameters with more hundreds of billions of training tokens, which verifies the scalability of the contributions.

**Reasons To Reject:**

* The 25% reduction in training compute seems rather small in the large picture. Many existing research (data engineering techniques, architectural changes, etc.) works have shown much greater compute savings. Moreover, this only reduces the training cost, but not the inference cost. So the total compute saving throughout the life of a model family is even lower when accounting for inference costs.

* It is more common to use different learning rates (LR) for different model sizes. Thus, the choice of using the same LR of 3e-4 for all models (mentioned in Line 117) is unreasonable. This also makes Section 4.4 unreasonable, since the proposed maximum LR is the more common practice in the community. This also means that the “independent” baseline reported in Table 3 and 8 is unreasonable, because it should have adjusted its maximum LR, which likely results in better performance.

* While the arguments regarding the potential effectiveness of model consistency (in Section 5.1) are convincing, the authors should have performed some experiments to show the actual quantitative benefits we can gain from employing some of the techniques mentioned (e.g., speculative decoding). It is very difficult to understand the significance of the reduction in KL divergence between different model sizes just from the numbers in Table 4.

---

> ### Author Response · Authors · 2025-06-02
>
> Thank you for your review. We would like to address your concerns.
>
> ## Weakness 1.
>
> > The 25% reduction in training compute seems rather small in the large picture. Many existing research (data engineering techniques, architectural changes, etc.) works have shown much greater compute savings. Moreover, this only reduces the training cost, but not the inference cost. So the total compute saving throughout the life of a model family is even lower when accounting for inference costs.
>
> Thank you for your feedback. On the impact of 25% savings, we respectfully disagree with the reviewer for the following two reasons:
>
> (1)  25% is not small in the development of LLM
> While a 25% reduction in training compute might initially seem modest compared to some other optimization techniques, its practical and financial significance in the context of large-scale language model (LLM) development is substantial. For instance, the 25% compute savings achieved in our experiments, equivalent to 3,200 GPU hours, translates to a direct cost reduction of approximately $44K USD (calculated using H100 GCP pricing https://cloud.google.com/compute/gpus-pricing ). This is a considerable sum that can impact project budgets.
> Furthermore, when considering the application of our progressive training to even larger, industry-standard model families like Llama 2 (e.g., 7B, 13B, 70B parameters) under a Chinchilla-optimal setting, the potential savings could escalate to an estimated $154K USD. This level of cost reduction is not trivial; it can make large-scale model development more accessible and sustainable, and our method has the potential to save large budgets and prevent wasting unnecessary computational resources.
>
> (2) Progressive training is complementary to existing techniques
> We believe that the value of our progressive training technique should not be judged solely by comparing its cost reduction percentage with that of orthogonal approaches, especially considering its complementarity.
> Regarding inference costs, while not a direct focus of this study, the procedural construction of a model family might offer benefits in selecting optimally sized models for diverse inference scenarios, potentially leading to indirect savings.

---

> > ### Author Response · Authors · 2025-06-02
> >
> > ## Weakness 2.
> > > It is more common to use different learning rates (LR) for different model sizes. Thus, the choice of using the same LR of 3e-4 for all models (mentioned in Line 117) is unreasonable. This also makes Section 4.4 unreasonable, since the proposed maximum LR is the more common practice in the community. This also means that the “independent” baseline reported in Table 3 and 8 is unreasonable, because it should have adjusted its maximum LR, which likely results in better performance.
> >
> > Thank you for your feedback. We appreciate the opportunity to discuss our learning rate (LR) choices.
> > The LR of 3×10−4 used for our "Independent" baseline (Tables 1 & 2 in the manuscript) is a widely adopted and reasonable setting for pre-training large language models.
> > As we note in the paper, larger models can be sensitive to LR, often requiring conservative choices to avoid instability like loss spikes. For instance, the Llama 1 & 2 models used this exact 3×10−4 LR for most sizes, reducing it to 1.5×10−4 only for their largest 70B variant, presumably for stability reasons [1].
> > Our initial fixed LR setup reflects this common and sometimes necessary conservative practice, providing a solid foundation for comparison.
> > While our initial fixed LR is a standard and justifiable approach, we agree that exploring LR adjustments tailored to model size, as discussed in Section 4.4 of the manuscript, is a valuable investigation.
> > To provide the comprehensive comparison you suggested for Table 3, we present results for an "Independent" baseline that also uses an adjusted LR schedule, identical to our "Progressive with LR adjustment" models.
> > Below is a comparison of the 2x Chinchilla law setting with adjusted learning rates, similar to Table 3 in the manuscript, showing 2B and 4B models:
> >
> > | Model Size | Training Method | Adjusted LR | Valid Perplexity ↓ | Wikitext Perplexity ↓ | LAMBADA Acc ↑ | ARC-e Acc ↑ | ARC-c Acc ↑ | Winogrande Acc ↑ | PIQA Acc ↑ | OBQA Acc ↑ | HellaSwag Acc ↑ |
> > |------------|-----------------|---------------|--------------------|-----------------------|---------------|-------------|-------------|------------------|------------|------------|-----------------|
> > | 2B         | Independent     | 1.1e-3        | 10.11              | 16.17                 | 49.4          | **72.6** | **42.9** | 59.4             | **75.3** | **41.6** | 61.3            |
> > |            | Progressive     | 1.1e-3        | **10.07** | **16.00** | **52.6** | 71.8        | 42.2        | **61.9** | **75.3** | 41.4       | **62.5** |
> > | 4B         | Independent     | 7e-4          | 8.77               | **12.98** | 54.2          | 74.3        | 45.0        | 62.4             | 76.6       | 40.8       | 66.7            |
> > |            | Progressive     | 7e-4          | **8.72** | 13.63                 | **55.7** | **74.5** | **47.8** | **65.2** | **77.4** | **43.0** | **68.1** |
> >
> > These results underscore that even when compared against an independently trained baseline with a similarly adjusted LR schedule, our progressive training method generally maintains comparable or superior performance. This is in addition to its primary contribution of significant FLOPs reduction.
> >
> > ### References.
> > [1] Touvron et al. 2023. Llama 2: Open Foundation and Fine-Tuned Chat Models.

---

> > ### Author Response · Authors · 2025-06-02
> >
> > ## Weakness 3.
> > > While the arguments regarding the potential effectiveness of model consistency (in Section 5.1) are convincing, the authors should have performed some experiments to show the actual quantitative benefits we can gain from employing some of the techniques mentioned (e.g., speculative decoding). It is very difficult to understand the significance of the reduction in KL divergence between different model sizes just from the numbers in Table 4.
> >
> > We thank the reviewer for highlighting the need to experimentally validate the actual quantitative benefits of model consistency (Section 5.1), particularly for speculative decoding. To address this, we conducted new experiments.
> >
> > **Speculative Decoding Experiment:**
> > - Setup:
> >   - 8B generator model, 4B drafter model
> >   -  K=8 speculative tokens
> >   - 1,000 prompts from FineWeb-Edu validation.
> > - Metrics
> >   - $D_{KL}(P_{\mathrm{drafter}} \textbar \textbar P_{\mathrm{generator}})$: KL divergence between the draft and generator model's output distributions, calculated on the FineWeb-Edu validation set (lower indicates higher consistency) as noted in Section 5.
> >   - Acceptance Rate (%): Average percentage of tokens drafted by the 4B model accepted by the 8B model as evaluated in the original speculative decoding paper [1].
> >   - Generation Time (seconds): Average generation time for all prompts.
> > - Comparison: To specifically isolate how the draft model's characteristics, particularly its consistency with the generator, affect speculative decoding performance, we fixed the generator model as the 8B Progressive model. We then compared outcomes when using 4B draft models constructed via Progressive versus Independent training approaches.
> >
> > - Results:
> > | Draft Model (4B) | Generator Model (8B) | $D_{KL}(P_{\mathrm{drafter}} \textbar \textbar  P_{\mathrm{generator}})$ | Acceptance Rate (%) | Generation Time (s) |
> > | :--------------- | :------------------- | :------------------------------------------------- | :------------------ | :-------------------- |
> > | Progressive | Progressive | 0.3378 | 93.20 | 7.02 |
> > | Independent | Progressive | 0.4281 | 87.2 | 8.24 |
> >
> > - Analysis:
> > The experimental results clearly show the quantitative benefits of using a draft model from Progressive Training for speculative decoding. Notably, the Progressive-Progressive (Prog-Prog) pairing achieved a 14.8% faster inference time compared to an independently trained drafter (Inde-Prog), a real-world efficiency gain, stemming from better model consistency via Progressive Training.
> >
> >
> > ### References.
> > [1] Leviathan et al. ICML 2023. Fast Inference from Transformers via Speculative Decoding.

---

> > ### Author Response · Authors · 2025-06-02
> >
> > We would like to address the questions as follows:
> >
> > ## Answer to *Why does using repeated data outperform the model that uses unique data (shown in Table 2)?*
> > We clarify that Progressive+Fixed Data achieves comparable, not superior, performance to the unique data approach (Progressive). The slight performance variations are expected due to different data ordering and sampling during training.
> > This comparable performance is consistent with prior work showing that training with repeated data up to ~4 epochs maintains similar performance without significant degradation [1]. In our Progressive+Fixed Data setting, we reuse 140B tokens (less than 1.5 epochs of repetition), which falls well within the range where data repetition doesn't harm performance.
> >
> > ## Answer to *Can we perform progressive training in a more fine-grained manner? For instance, start from 1M parameters and progressively scale up to 8B?*
> >
> > Yes, progressive training can indeed be performed in a more fine-grained manner, though the practical benefits vary depending on the starting scale.
> >
> > **Technical Feasibility.**
> >
> > Progressive training is technically feasible across a wide range of model sizes. However, starting from extremely small scales (e.g., 1M parameters) provides diminishing practical benefits, as even foundational small models like GPT-2 [2] and BERT [3] typically start around 100M parameters.
> >
> > **Experimental Validation.**
> >
> > To validate the effectiveness of progressive training from smaller models, we conducted additional experiments with two smaller model families:
> > [100M, 200M,  400M, 800M] and [250M, 500M, 1B, 2B].
> > Results
> > |  |  | Perplexity ↓  | | Accuracy ↑  | |  |  |  | |  |
> > |-----|--|--|-|----|-|----------------|-|-------------|--|---|
> > | |    | Valid| Wikitext| LAMBADA| ARC-e  | ARC-c| Winogrande | PIQA | OBQA | HellaSwag   |
> > | 100 M | Independent | 24.53    | 56.3118    | 0.1939          | 0.4354         | 0.2602         | 0.5004            | 0.6034      | 0.308       | 0.2981      |
> > | 200 M       | Independent | 19.66         | 41.1199               | 0.2544          | 0.4823         | 0.2611         | 0.5091            | **0.6344** | **0.318** | 0.3312      |
> > | 200 M       | Progressive | **19.29** | **40.1259** | **0.2560** | **0.4874** | **0.2696** | **0.5201** | 0.6333      | 0.304       | **0.3395** |
> > | 400 M       | Independent | 16.3         | 31.3596               | 0.3082          | 0.5139         | **0.2856** | 0.5214            | 0.6703      | 0.324       | 0.3811      |
> > | 400 M       | Progressive | **15.29** | **30.8099** | **0.3219** | **0.5383** | 0.2833         | **0.5225** | **0.6730** | **0.328** | **0.3939** |
> > | 800 M       | Independent | 13.87         | 24.9751               | 0.3648          | **0.5888** | 0.2961         | 0.5146            | **0.6986** | 0.340       | 0.4479      |
> > | 800 M       | Progressive | **13.7** | **24.7707** | **0.3679** | 0.5791         | **0.3123** | **0.5343** | 0.6937      | **0.372** | **0.4598** |
> >
> > |   | | Perplexity ↓  |   | Accuracy ↑      |   |   |  |   |  | |
> > |--|-------------|---------------|----|-----------------|---|---|---|-------------|-------------|-------------|
> > |             |             | Valid         | Wikitext              | LAMBADA         | ARC-e          | ARC-c          | Winogrande        | PIQA        | OBQA        | HellaSwag   |
> > | 250 M       | Independent | 18.88         | 36.6648               | 0.2633          | 0.4975         | 0.2611         | 0.5051            | 0.636       | 0.302       | 0.3433      |
> > | 500 M       | Independent | 15.97         | 29.2741               | 0.315           | 0.5257         | **0.2858** | 0.5099            | **0.6632** | **0.338** | 0.3931      |
> > | 500 M       | Progressive | **15.27** | **28.5509** | **0.3258** | **0.5501** | 0.2705         | **0.5122** | 0.6621      | 0.322       | **0.4047** |
> > | 1B      | Independent | 13.34         | 23.4164               | 0.3716          | 0.5867         | 0.3046         | 0.5075            | **0.697** | **0.358** | 0.4596      |
> > | 1B      | Progressive | **13.16** | **23.1414** | **0.379** | **0.6069** | **0.3191** | **0.5399** | 0.6948      | 0.354       | **0.4819** |
> > | 2B      | Independent | 11.50         | 19.2137               | 0.4262          | **0.6604** | **0.3746** | 0.5406            | **0.7291** | 0.388       | 0.5421      |
> > | 2B      | Progressive | **11.37** | **18.8873** | **0.4394** | 0.6347         | 0.366          | **0.5659** | 0.7236      | **0.392** | **0.5518** |
> >
> > The results above demonstrate that progressive training maintains its effectiveness even at smaller scales while requiring computational cost equivalent to training only the largest model in each family.
> >
> > ### References
> > [1] Muennighoff et al. NeurIPS 2023. Scaling Data-Constrained Language Models.
> >
> > [2] Radford et al. 2019. Language Models are Unsupervised Multitask Learners.
> >
> > [3] Devlin et al. NAACL 2019. BERT: Pre-training of Deep Bidirectional Transformers for Language Understanding.

---

> > > ### Author Response · Authors · 2025-06-02
> > >
> > > ## Answer to *Is the expansion method compatible with muP, which has become a popular technique in LLM training?*
> > > Thank you for this insightful question.
> > >
> > > The compatibility between Progressive Training and muP is an important consideration.
> > >
> > > **Theoretical Considerations.**
> > >
> > > The key challenge lies in whether model expansion methods can preserve the parameter scaling relationships that muP [1] requires. Recent theoretical work [2] suggests that Depth scaling is more naturally compatible with muP than width scaling, which requires careful coordination of initialization variance and learning rates.
> > >
> > > Since our progressive training uses bert2BERT (which expands both width and depth), this raises questions about maintaining muP's scaling laws during the expansion process. However, depth-wise model expansion (such as stacking [3] )may be more naturally compatible with muP principles.
> > >
> > > To explore this compatibility, we conducted a small-scale experiment using depth-wise stacking (which should be more muP-compatible) for progressive training.
> > >
> > > **Experiment.**
> > >
> > > - Model family: [500M (15 layers), 1B ( 30 layers ), 2B (60 layers)]
> > > - Learning rates tested: 1e-3, 2e-3, 4e-3, 8e-3 to assess transferability
> > > - Expansion method: Layer stacking only (no width changes)
> > > - Data setting: Chinchilla setting
> > > - Evaluation: Validation Perplexity
> > >
> > > The 500M model, when trained independently, achieved its best validation loss with a learning rate of 2e-3. This optimally performing 500M model served as the foundation for our progressive training.
> > > Specifically:
> > >
> > > - The 1B progressive model was initialized from this 500M independent model (which was trained with an LR of 2e-3)
> > > - Subsequently, the 2B progressive model was initialized from the 1B progressive model (which had been trained with an LR of 2e-3)
> > >
> > > For a comprehensive assessment of learning rate transferability and to identify optimal learning rates, the performance of these progressively trained models (1B and 2B) was then evaluated across the full range of tested learning rates (1e-3, 2e-3, 4e-3, 8e-3), as detailed in the results table.
> > >
> > > **Results.**
> > >
> > > | Model | Method      | Learning Rate vs Validation Loss             |
> > > | :---- | :---------- | :------------------------------------------- |
> > > | 500M  | Independent | 1e-3: 2.74, 2e-3: 2.72, 4e-3: 2.72, 8e-3: 2.74 |
> > > | 1B    | Independent | 1e-3: 2.59, 2e-3: 2.56, 4e-3: 2.55, 8e-3: 2.60 |
> > > | 1B    | Progressive | 1e-3: 2.51, 2e-3: 2.50, 4e-3: 2.51, 8e-3: 2.55 |
> > > | 2B    | Independent | 1e-3: 2.44, 2e-3: 2.40, 4e-3: 2.40, 8e-3: 2.44 |
> > > | 2B    | Progressive | 1e-3: 2.32, 2e-2: 2.32, 4e-3: 2.33, 8e-3: 2.36 |
> > >
> > > The table above shows that progressive training consistently outperforms independent training across all model sizes while maintaining reasonable learning rate transferability. The optimal learning rate ranges remain within 1e-3 to 4e-3 for both approaches, suggesting that depth-wise expansion preserves the scaling relationships necessary for muP compatibility. Although there is some shift in the precise optimal learning rates, the overall stability indicates that progressive training with depth-wise stacking can work harmoniously with muP principles. Future work should investigate the other model expansion method's compatibility with muP at larger scales.
> > >
> > > ## Answer to *I think it is more common to have a space between the number before “ZFLOPs” in, for instance, “20.4ZFLOPs”*
> > >
> > > Thank you for pointing this out. We will revise the notation throughout the paper.
> > >
> > > ### References.
> > > [1] Yang et al. NeurIPS 2021. Tensor Programs V: Tuning Large Neural Networks via Zero-Shot Hyperparameter Transfer.
> > >
> > > [2] Day et al. 2025. Don't be lazy: CompleteP enables compute-efficient deep transformers.
> > >
> > > [3] Du et al. NeurIPS 2024. Stacking Your Transformers: A Closer Look at Model Growth for Efficient LLM Pre-Training.

---

> > > > ### Author Response · Authors · 2025-06-10
> > > >
> > > > Thank you for increasing your score. With the discussion period nearing its conclusion,  we would appreciate it if you could let us know if you still have any concerns.

---

### Official Review · Reviewer_hRL8 · 2025-05-13

**Rating:** 3
**Confidence:** 3
**Ethics Flag:** 1

**Summary:**

This paper presents a method for constructing a family of language models using progressive training through model expansion. The authors claim a 25% reduction in computational cost compared to training each model independently, while maintaining comparable performance. Additional benefits include consistent behavior across different model sizes.

**Reasons To Accept:**

- The problem of efficiently training a model family is relevant and timely, particularly with increasing deployment of LLMs across diverse platforms and constraints.
- The proposed progressive training pipeline is sensible and leverages model expansion effectively.
- The method shows empirical improvements over independent training baseline in terms of both computational cost and performance on downstream tasks.

**Reasons To Reject:**

- The novelty of paper is limited. The method employs existing off-the-shelf model expansion tools. The idea of progressive training is also not new. The paper essentially repackages them for constructing a model family without proposing a fundamentally new algorithm or mechanism.
- The most interesting aspect is the problem definition and the constraints on the FLOP computations and the token budgets. However, this is not algorithmically addressed. In the experiments, instead of searching or optimizing for configurations under constraints, the authors use a predefined token budget schedule. This makes the paper pure empirical without any new proposed methods or surprising findings.
- Section 5.1 provides an extended discussion on the advantages of consistency across model sizes, such as in speculative decoding. However, these claims are mostly speculative. While KL divergence analysis (Section 5.2) does quantify similarity across adjacent models, the claimed practical benefits are not experimentally verified.

---

> ### Author Response · Authors · 2025-06-02
>
> We thank the reviewer for reading our paper and feedback. We would like to address the weaknesses as follows:
>
> ## Weakness 1.
> >  The novelty of paper is limited. The method employs existing off-the-shelf model expansion tools. The idea of progressive training is also not new. The paper essentially repackages them for constructing a model family without proposing a fundamentally new algorithm or mechanism.
>
>
> Our primary contribution lies in proposing a comprehensive learning framework for effectively constructing a model family and empirically demonstrating its effectiveness.
> While the reviewer appears to have concluded that our work lacks novelty because we employed an existing method for the model expansion part, we believe that the novelty of our work is in line with COLM's recognition of contributions in **Empiricism, Data, and Evaluation**, **Technological Impact**, rather than being limited to algorithmic innovation (https://colmweb.org/ReviewGuide.html).
>
> Our paper presents novel empirical findings specific to this procedural approach to model family construction.
> For example, we demonstrate that our iterative expansion procedure using existing methods leads to greater behavioral consistency across model sizes, as quantified by KL divergence between adjacent models in the family.
> Crucially, we also show that strategically adjusting learning rates based on model size during this progressive family construction yields more efficient training and better-performing model families.
> These empirical insights into building model families with existing tools offer new, practical contributions to the field.
>
>
>
> ## Weakness 2.
> > The most interesting aspect is the problem definition and the constraints on the FLOP computations and the token budgets. However, this is not algorithmically addressed. In the experiments, instead of searching or optimizing for configurations under constraints, the authors use a predefined token budget schedule. This makes the paper pure empirical without any new proposed methods or surprising findings.
>
> Our research objective is to efficiently construct model families within given computational budgets that reflect standard and practical settings.
> We tackle this objective under widely recognized resource constraints (Chinchilla law and 2x Chinchilla law), which represent standard LLM pre-training research practices rather than arbitrary choices [1][2].
> Our experimental design comprehensively addresses the core research question through controlled comparisons that isolate the effects of progressive training.
> While empirically focused, our methodology is both rigorous and complete for the stated objectives.
> We would also like to emphasize that pretraining demands substantial computational resources and is time-intensive.
> As such, we generally refrain from using complex methods during pretraining in order to ensure fast and efficient training.
> For this reason, we adopted a simple, predefined scheduling strategy.
> We believe that most researchers and developers with experience in pretraining would agree with this point of view.
> Furthermore, our findings are not trivial: (1) computational efficiency with maintained performance, (2) behavioral consistency across model families with practical implications for techniques like speculative decoding, and (3) empirical validation that benefits persist when controlling for data exposure.
> These represent meaningful advances in practical model family construction.
>
> ### References.
> [1] Lialin et.al. ICLR 2024. ReLoRA: High-Rank Training Through Low-Rank Updates.
>
> [2] Zhao et al. ICML 2024. GaLore: Memory-Efficient LLM Training by Gradient Low-Rank Projection.

---

> > ### Comment · Reviewer_hRL8 · 2025-06-09
> >
> > Thank you to the authors for the detailed response and for providing additional experimental results. I agree that not all papers must introduce substantial algorithmic novelty. As the authors noted, valuable contributions can also lie in areas such as empiricism, data and evaluation, and technological impact. However, these types of contributions demand a rigorous and well-designed experiment.
> >
> > While I acknowledge the computational constraints, the experimental evaluation remains quite limited. As a result, the findings appear neither novel nor particularly surprising. For these reasons, I will retain my current score.

---

> > > ### Author Response · Authors · 2025-06-10
> > >
> > > Thank you for your response. While we respect the reviewer's concern that the findings don't seem surprising, we believe our empirical contributions represent advances in practical model family construction. Beyond our main contribution of achieving computational savings for model family construction, our work introduces two additional novel findings: Model Consistency introducing KL divergence-based analysis across model families, which reveals that progressive training produces significantly more coherent model behaviors compared to independent training; and Empirically Validated Speculative Decoding demonstrating concrete inference speedup with improved acceptance rates, directly validating the practical benefits of model family consistency for accelerated inference applications. We believe these overall findings represent a meaningful contribution to the community.
> > >
> > > Additionally, we note the reviewer's assessment that "the experimental evaluation remains quite limited" and that "well-designed experiments" are needed for valuable empirical contributions. However, we respectfully observe that the original three "Reasons to Reject" did not include specific concerns about experimental design or methodology. While we remain committed to rigorous empirical work, we are uncertain what specific additional experimental directions would address this concern, given the comprehensive validation we have provided.
> > >
> > > Furthermore, we have systematically addressed each of the three "Reasons to Reject" through clarifications and additional experimental evidence, and therefore believe the concerns raised have been resolved. We respectfully note a potential inconsistency between the retained score of 3 ("Clear rejection") and the reviewer's acknowledgment in "Reasons to Accept" that our method "shows empirical improvements over the independent training baseline in terms of both computational cost and performance on downstream tasks." Given this recognized contribution and the additional validation we have provided, we would appreciate the reviewer's reconsideration of the score.

---

> ### Author Response · Authors · 2025-06-02
>
> ## Weakness 3.
> > Section 5.1 provides an extended discussion on the advantages of consistency across model sizes, such as in speculative decoding. However, these claims are mostly speculative. While KL divergence analysis (Section 5.2) does quantify similarity across adjacent models, the claimed practical benefits are not experimentally verified.
>
> We thank the reviewer for highlighting the need to experimentally validate the claimed benefits of model consistency (Section 5.1), particularly for speculative decoding. To address this, we conducted new experiments.
>
> **Speculative Decoding Experiment:**
> - Setup:
>   - 8B generator model, 4B drafter model
>   -  K=8 speculative tokens
>   - 1,000 prompts from FineWeb-Edu validation.
> - Metrics
>   - $D_{KL}(P_{\mathrm{drafter}} \textbar \textbar P_{\mathrm{generator}})$: KL divergence between the draft and generator model's output distributions, calculated on the FineWeb-Edu validation set (lower indicates higher consistency) as noted in Section 5.
>   - Acceptance Rate (%): Average percentage of tokens drafted by the 4B model accepted by the 8B model as evaluated in the original speculative decoding paper [1].
>   - Generation Time (seconds): Average generation time for all prompts.
> - Comparison: To specifically isolate how the draft model's characteristics, particularly its consistency with the generator, affect speculative decoding performance, we fixed the generator model as the 8B Progressive model. We then compared outcomes when using 4B draft models constructed via Progressive versus Independent training approaches.
>
> - Results:
> | Draft Model (4B) | Generator Model (8B) | $D_{KL}(P_{\mathrm{drafter}} \textbar \textbar  P_{\mathrm{generator}})$ | Acceptance Rate (%) | Generation Time (s) |
> | :--------------- | :------------------- | :------------------------------------------------- | :------------------ | :-------------------- |
> | Progressive | Progressive | 0.3378 | 93.20 | 7.02 |
> | Independent | Progressive | 0.4281 | 87.2 | 8.24 |
>
> - Analysis:
>
> The results clearly demonstrate the practical advantages of employing a draft model constructed through Progressive Training. The pairing of a 4B Progressive draft model with an 8B Progressive generator model (Prog-Prog) exhibited superior performance, underpinned by greater consistency between the two models. This enhanced consistency is quantified by a lower $D_{KL}(P_{\mathrm{drafter}} \textbar \textbar  P_{\mathrm{generator}})$ value for the Prog-Prog pair, compared to using an independently trained model as a drafter. This improved alignment in output distributions directly translated to more effective speculative decoding: the Prog-Prog configuration achieved a significantly higher acceptance rate of 93.20%, as opposed to 87.2% for the Inde-Prog setup. Consequently, this led to a reduction in inference time, with the Prog-Prog pair completing generation approximately 14.8% faster (7.02s vs. 8.24s). These findings experimentally substantiate our claim in Section 5.1 that the consistency fostered by Progressive Training offers tangible benefits. The lower KL divergence between progressively trained models leads to more effective draft proposals in speculative decoding, thereby improving acceptance rates and reducing inference latency. This highlights that Progressive Training not only provides an efficient way to construct model families but also yields models that are better suited for downstream applications like speculative decoding due to their enhanced inter-model consistency.
>
> ### References.
> [1] Leviathan et al. ICML 2023. Fast Inference from Transformers via Speculative Decoding.

---

### Decision · Program_Chairs · 2025-07-08

**Decision:**

Accept

**Comment:**

This work presents a method for progressive training a family of language models—that is, gradually expand the model as it trains to obtain multiple complete language models in a signle training run. The paper experiments with a model family of 1B to 8B parameters; results show that the proposed model achieves same results of independently trained models while achieving a 25% reduction in compute budget.

All reviews agree that this work is very relevant and timely; they also remark the soundness of the proposed method, as well as the empirical evidence provided. Some also praise the ease of reproducibility, asn well as the writing style.

Reviewers raise three main concerns with this work: lack of novelty, some experimental configurations (namely, LR tuning), as well as claims in sections 5.2 not being supported by experimental evidence. During rebuttal, authors address the last two with additional experimental evidence. With respect of lack of novelty, I think the ample experimental provided by authors more than make up for it, as it makes this method considerably easier to adopt by practicioners.

Authors should make sure to incorporate ALL aditional experiments and reference provided by authors in their final camera ready.

### Pros

- topic of efficient pretraining of a family of LLM is timely and important (hRL8, D8FB, Az33)
- the proposed pipeline is sound (hRL8, D8FB)
- empirical evidence of proposed method (hRL8, D8FB, Az33)
- paper well written (D8FB)
- work is easy to reproduce (oPhZ)
- -25% is significant reduction in compute hours (oPhZ)
- more effective speculative decoding  pipeline (oPhZ)

### Cons

- limited novelty: off-the-shelf methods, and progressive training is not new (hRL8, oPhZ, Az33)
- training configurations are arbitrarily chosen, rather justified by experiments or theoretical framework (hRL8)
- authors do not provide empirical evidence that progressively constructed model family helps with speculative decoding (hRL8, D8FB)
  - **addressed in rebuttal**
- -25% is limited reduction in compute costs (D8FB)
- LR is not tuned per model configuration (D8FB)
  - **addressed in rebuttal**
- speculative decoding run sees 2x the epochs (oPhZ)
- no robustness or safety benchmakrs (oPhZ)
- some related work missing (oPhZ)
- intuition behind the proposed method is unclear (Az33)
- analysis limited to 8B models; unclear how much the approach would scale past that (oPhZ,Az33)